# Potential Biomarkers and Endometrial Immune Microenvironment in Recurrent Implantation Failure

**DOI:** 10.3390/biom13030406

**Published:** 2023-02-21

**Authors:** Fangfang Li, Wenxin Gao, Yanmei Li, Yiqing Wang, Lin Liu, Xuehong Zhang

**Affiliations:** 1The First School of Clinical Medicine, Lanzhou University, Lanzhou 730000, China; 2Gansu International Scientific and Technological Cooperation Base of Reproductive Medicine Transformation Application, Key Laboratory for Reproductive Medicine and Embryo of Gansu, Lanzhou 730000, China

**Keywords:** recurrent implantation failure, biomarker, immune cell infiltration, immune pathway, lipid metabolism

## Abstract

The molecular mechanisms underlying unexplained recurrent implantation failure (RIF) remain unclear. This study aimed at identifying potential biomarkers, exploring relevant signaling pathways, and analyzing the contribution of immune cell infiltration in RIF. Microarray expression datasets were extracted from the Gene Expression Omnibus database to perform bioinformatic analyses. The results showed that ten hub genes may predict RIF with high specificity and sensitivity (area under the curve = 1.000). Protein–protein interaction analysis revealed close interactions between the hub genes and the endometrial receptivity array. The real-time quantitative polymerase chain reaction further validated three potential biomarkers (*RAB32, TRIB2,* and *FAM155B*). Functional enrichment analyses indicated that immune pathways were significantly downregulated and lipid metabolism pathways were significantly upregulated in RIF compared with the controls. Significant negative correlations were observed between fatty acid biosynthesis and the immune pathways. Immune cell infiltration, including those in CD56dim natural killer, dendritic, Th1, Th2, and regulatory T cells, as well as macrophages, was significantly reduced in RIF compared with the controls used herein. This study may provide a novel perspective on the diagnosis and treatment of RIF.

## 1. Introduction

With advancements in assisted reproductive technology, the success rate per cycle has increased remarkably. However, recurrent implantation failure (RIF) still affects 10–15% of couples undergoing in vitro fertilization (IVF) or intracytoplasmic sperm injection (ICSI) treatment, and the causes underlying up to 50% of these cases remain elusive [1,2,3]. The complex implantation process is divided into three distinct phases: apposition, attachment, and invasion [3,4,5,6]. Apposition occurs when blastocyst L-selectins interact with corresponding ligands from the receptive endometrium. Attachment refers to the interaction between integrins from the luminal epithelial lining and trophoblasts. In the final invasion phase, trophoblast cells proliferate, differentiate, penetrate the uterine epithelium, invade the underlying endometrial stroma, and stimulate decidual transformation, the endpoint of which is spiral artery remodeling and placentation. Successful implantation relies on the receptive endometrium and a developmentally competent blastocyst.

Although the pathogenesis of unexplained RIF remains controversial, immune cells and immune-related pathways are increasingly recognized for their importance in establishing receptivity and initiating pregnancy. During embryo implantation, immune cells play tissue remodeling and immunomodulatory roles, including promoting epithelial attachment, modulating trophoblast invasion, regulating decidual transformation, remodeling uterine vasculature, and controlling inflammatory activation, along with suppressing inappropriate immunity to paternal human leukocyte antigen (HLA) [4,7,8]. Women with RIF may experience changes in the number or function of endometrial immune cells. Innate immune cells, particularly macrophages [9], dendritic cells (DCs) [10], and uterine natural killer (uNK) cells [11], influence endometrial receptivity and trophoblast invasion by releasing cytokines and promoting vascular adaptation and immunoregulation. uNK cells release granulocyte-macrophage colony-stimulating factor (GM-CSF), vascular endothelial growth factor (VEGF), placental growth factor, interleukin 8 (IL-8), interferon γ (IFN-γ), and tumor necrosis factor (TNF) to regulate trophoblast invasion and maternal vascular adaptation [12]. Macrophages release leukemia inhibitory factors to modify the glycosylated structures of epithelial cells required for embryo attachment [13]; macrophages facilitate implantation by regulating corpus luteum development and progesterone secretion [14]. Furthermore, balances between effector T cells (Teffs) and regulatory T cells (Tregs) control maternal immune tolerance to the fetus [15]. In general, immune cell populations work together to conduct maternal-fetal communication. The immune response is also affected by metabolism, nutrition, inflammation, autoimmunity, and age [16,17,18,19].

Uterine immune cells and molecules are essential elements of endometrial biology. Their complex interactive networks during implantation, fluctuations in abundance and phenotypes within individuals from one cycle to another, and high variability between individuals suggest that they are gatekeepers of the receptive endometrium [4]. Nevertheless, the molecular mechanisms underlying the pathogenesis of unexplained RIF remain unclear. The complexity within immune signaling networks makes the identification of a limited number of genes predicting endometrial receptivity difficult.

In recent years, microarray technology and integrated bioinformatic analyses have been employed to identify novel diagnostic biomarkers. Herein, based on the screening of differentially expressed genes (DEGs) and weighted gene co-expression network analysis (WGCNA), three machine-learning algorithms were utilized to sieve core biomarkers in unexplained RIF, followed by the use of a receiver operating characteristic curve (ROC) and a diagnostic nomogram to construct a diagnostic prediction model. Real-time quantitative polymerase chain reaction (RT-qPCR) was conducted to further validate hub genes. Gene Ontology (GO), Kyoto Encyclopedia of Genes and Genomes (KEGG), Gene Set Enrichment Analysis (GSEA), and single-gene GSEA were used to analyze the signaling pathways. A protein–protein interaction (PPI) network was constructed to analyze interactions between hub genes and 238 endometrial receptivity array (ERA) genes. The infiltration of 28 immune cells [20] in unexplained RIF was analyzed herein using single-sample GSEA (ssGSEA) to gain an in-depth understanding of the pathogenesis of unexplained RIF and bring new perspectives to its research and treatment.

## 2. Materials and Methods

### 2.1. Data Collection and Extraction

Microarray expression data for RIF were extracted from the Gene Expression Omnibus (GEO, http://www.ncbi.nlm.nih.gov/geo/, accessed on 14 May 2022) database (GSE111974 and GSE92324). The GSE111974 dataset was derived from the GPL17077 platform, with 24 RIF cases and 24 healthy controls in the natural cycle. The GSE92324 dataset was derived from the GPL10558 platform, with 10 RIF cases and 8 healthy controls in the controlled ovarian stimulation (COS) cycle. All samples from the two datasets were obtained from human endometrial tissues. Demographics and hormone levels between RIF patients and healthy controls are detailed in Appendix A, according to previously published articles on the two datasets [21,22].

### 2.2. Data Preprocessing and Identification of DEGs

Data normalization, probe annotation, and DEG analysis of the GSE111974 dataset were performed using the R-limma package, setting the DEG screening thresholds to adjusted *p* < 0.05 and *|log2 FoldChange (FC)|* > 1. 

### 2.3. Co-Expression Network Construction

Expression profile data for all genes were extracted from the GSE111974 dataset. The gene co-expression network was constructed using the R-WGCNA package: (1) the “goodSampleGenes” function was applied to check the integrity of the data; (2) matrix data were converted to an adjacency matrix, setting the soft-threshold β to 5 with the help of the “pickSoftThreshold” function; (3) adjacency matrix conversion to a topological overlap matrix (TOM) was implemented, clustering genes using 1-TOM as the distance and identifying modules using dynamic tree cuts; (4) similar modules were merged, and module eigengenes were clustered, setting the merging threshold to 0.25; and (5) modules were combined with clinical traits to analyze their correlations and p-values. Gene modules with significant clinical correlations were selected as the candidate hub modules.

### 2.4. Identifying Hub Genes for Predicting RIF

Genes from the candidate hub modules in WGCNA were intersected with the identified DEGs (GSE111974) to obtain intersecting genes with both differences and correlations. To screen hub genes of RIF in both natural and COS cycles, the intersecting genes in the natural cycle of the GSE111974 dataset were intersected with DEGs in the COS cycle of the GSE92324 dataset. After the preliminary screening, support vector machine recursive feature elimination (SVM-RFE), least absolute shrinkage and selection operator (LASSO), and random forest (RF) analyses were applied, with the help of R-e1071, R-glmnet, and R-randomForest packages, respectively. The number of top-ranked genes was identified based on the minimum error in the SVM-RFE and RF, with 10-fold cross validation, and the minimal binomial deviance in LASSO, with 10-fold cross validation. Overlapping genes from the three machine-learning algorithms were screened as hub genes. Spearman correlations between the hub genes were analyzed. The ROC curve was constructed to assess the diagnostic value of the hub genes and the levels of hub genes distinguishing between RIF and healthy controls using the R-pROC package. A diagnostic nomogram was used to construct the diagnostic prediction model using the R-rms package. The combined diagnosis of the hub genes was constructed using logistic regression, with 8-fold cross validation, in the GSE111974, and 3-fold cross validation in the GSE92324, using the Python-sklearn package. The result was visualized using the ROC curve with the Python-matplotlib package. The hub gene expression levels in RIF patients and healthy controls were evaluated using box plots.

### 2.5. Functional Enrichment Analysis

Based on the preliminary screened DEGs, GO and KEGG analyses were performed using the R package “clusterProfiler,” with a threshold *p* < 0.05. GSEA of the GSE111974 dataset allowed for further understanding of the biological signal pathways using the R package “clusterProfiler” with a threshold *p* < 0.05. ssGSEA for the GSE111974 dataset was conducted using the R package “gsva.” According to the results of ssGSEA, Spearman correlations between signaling pathways and between the signaling pathways and the hub genes were analyzed.

### 2.6. Prediction of the Underlying Mechanism of Hub Genes 

To explore the underlying mechanism of hub genes, single-gene GSEA in the GSE111974 dataset was used to find the significant enrichment of KEGG pathways using R package “clusterProfiler” with a threshold *p* < 0.05.

### 2.7. PPI and Correlation Analyses of Hub Genes and ERA

The ERA of 238 genes may be instructive in assessing receptive endometrium, identifying the window of implantation (WOI), and improving implantation rates in women with RIF [23]. The PPI network of the hub genes and ERA was constructed by the STRING database (http://string-db.org, accessed on 1 October 2022) to provide references for exploring the interaction between hub genes and endometrial receptivity, which provided a basis for studying the occurrence and progression of RIF. The PPI network was visualized by Cytoscape (version 3.9.1), with an interaction score of >0.15. All genes were ranked using betweenness centrality by the CytoNCA plugin. In addition, the Spearman correlation between the hub genes and the ERA was analyzed in the GSE111974 and GSE92324 datasets.

### 2.8. Assessment of Immune Cell Infiltration and Its Correlation with Hub Genes

The relative infiltration levels of 28 immune cells were quantified using the ssGSEA algorithm. A violin plot was constructed to demonstrate the differential infiltrating levels of 28 immune cells between RIF and healthy controls, and Spearman correlation plots among 28 immune infiltrating cells and between the immune cells and hub genes were analyzed.

### 2.9. Clinical Sample Collection

Experimental protocols were approved by the Ethics Committee of the First Hospital of Lanzhou University (No: LDYYLL2019-45). Four RIF patients and four controls were recruited from the first Hospital of Lanzhou University. Written informed consent was obtained from all participants. RIF patients underwent at least three failed embryo transfer cycles, with at least four high-quality embryos. The control women achieved clinical pregnancy after IVF/ICSI treatment due to fallopian tube or male infertility factors. Inclusion and exclusion criteria were as follows: (1) under the age of 40; (2) regular menstrual cycles (25–35 days); (3) normal basal serum sex hormone levels of follicle-stimulating hormone (FSH < 10 mIU/mL), luteinizing hormone (LH < 10 mIU/mL), and estradiol (E2 < 50 pg/mL); (4) exclusion due to steroid use within the previous 2 months, chromosomal abnormalities, uterine malformations (i.e., polyps, fibroids, and intrauterine adhesions), endocrine metabolic abnormalities (i.e., abnormal blood glucose levels, polycystic ovary syndrome, insulin resistance, thyroid dysfunction), and autoimmune diseases. The pipe suction curettage was performed on day LH + 7 to obtain endometrial tissues.

### 2.10. RT-qPCR

Total RNA from endometrial tissues was extracted using TRIzol regent (Invitrogen, Carlsbad, CA, USA) and then reverse-transcribed into cDNA using the PrimeScript RT Reagent Kit (Servicebio, Co., Ltd., Wuhan, China). RT-qPCR was performed to detect target mRNA expression (ABI 7900HT, Thermo Fisher Scientific, Waltham, MA, USA), which was normalized to *ACTB* expression. The RT-qPCR primer sequences are listed in Appendix A. The 2^−ΔΔ^Ct method was used to analyze the results.

### 2.11. Statistical Analysis

Statistical analysis was conducted using R (4.1.3), Python (3.8), or GraphPad Prism (8.0.1). *p* < 0.05 was considered statistically significant, unless a specific *p*-value was given.

## 3. Results

### 3.1. Co-Expression Network Construction and Hub Modules Identification

The gene co-expression network was established, based on the extracted expression profile data of all genes (GSE111974), followed by a sample dendrogram and trait heatmap (Appendix A). A scale-free network was constructed with a soft threshold of β = 5 (scale-free R2 = 0.9) (Appendix A). A hierarchical clustering tree was built using a dynamic hybrid cutting method, which generated 31 gene modules (Figure 1A) and module genes. The correlations between gene modules with RIF are presented as heatmap–module–trait relationships (Figure 1B). Significant correlations (*p* < 0.05) were observed in ten modules (dark slate blue, sienna3, plum2, royal blue, lightcyan1, black, blue, magenta, midnight blue, and grey60 modules), of which the magenta module demonstrated the highest correlation with RIF (correlation coefficient (cor) = 0.89; *p* = 2 × 10^−17^) (Figure 1B). Ten significantly correlated gene modules were selected as candidate hub modules for subsequent analysis.

### 3.2. Identification of DEGs and Screening of Hub Genes 

A total of 444 DEGs, including 253 upregulated and 191 downregulated genes, were obtained from the GSE111974 dataset (Figure 2A). A total of 19,796 module genes were identified from ten candidate hub modules in WGCNA. Next, 358 genes were screened, based on the intersection of DEGs (GSE111974) and 19,796 module genes (Figure 2B). Subsequently, 358 genes were further intersected with DEGs (GSE92324) to accomplish preliminary screening of the 85 intersecting genes (Figure 2C). Based on the 85 DEGs, the SVM-RFE (Figure 2D), LASSO (Figure 2E), and RF (Figure 2F) evaluations identified 47, 10, and 48 genes, respectively. Ten overlapping genes were screened as hub genes (Figure 2G). Correlations between the ten hub genes were significant (*p* < 0.01). *TRIB2* and *RAB32* were negatively correlated with *NRCAM, ATP13A4, SHROOM3, FAM155B, ZNF436, SLC25A29, PDZD8,* and *HOXB3* (Figure 2H). *RAB32* had the highest negative correlation with *ZNF436* (cor = −0.81), *SLC25A29* (cor = −0.80), and *PDZD8* (cor = −0.80). *FAM155B* had the highest positive correlation with *SLC25A29* (cor = 0.82). 

### 3.3. Functional Enrichment Analysis

GO analysis of the 85 DEGs was conducted to understand the biological processes, cellular components, and molecular functions associated with RIF (Figure 3A). The biological processes mainly involve immune and inflammatory processes (the nucleotide-binding domain, leucine-rich repeat-containing receptor signaling pathway, and nucleotide-binding oligomerization domain-containing signaling pathway) and substance metabolism and transport (carboxylic acid transmembrane transport, sterol metabolic process, and pigment accumulation), which are closely associated with the hub genes (Figure 3B). The cellular components exhibited apical parts of the cell and an apical plasma membrane. Molecular functions included cholesterol binding, NADP binding, oxidoreductase activity, and sterol binding. KEGG analysis showed that the 85 DEGs were enriched in immune and inflammatory pathways (nuclear factor-kappa B (NF-κB) and TNF signaling pathways), metabolic pathways (adipocytokine signaling pathway, nucleotide metabolism, and drug metabolism-cytochrome P450), and alcoholic liver disease associated with lipid metabolism disorders and inflammatory responses (Figure 3C). The immune and inflammatory pathways and adipocytokine signaling pathways were closely linked (Figure 3D).

To explore the changes in immune, inflammatory, and metabolic pathways, the GSEA of the GSE111974 dataset was performed. The results demonstrated that lipid metabolism pathways (i.e., fatty acid biosynthesis, ovarian steroidogenesis, and steroid hormone biosynthesis) were significantly upregulated in RIF (*p* < 0.05), and fatty acid biosynthesis had the highest enrichment score among the upregulated pathways (enrichment score = 0.616, *p* = 0.009) (Appendix A). By contrast, the immune and inflammatory pathways (i.e., graft-versus-host disease, allograft rejection, antigen processing and presentation, the chemokine signaling pathway, the C-type lectin receptor signaling pathway, Th1 and Th2 cell differentiation, Th17 cell differentiation, and the T cell receptor signaling pathway) were significantly downregulated (*p* < 0.05), accounting for nearly one-third of all downregulated pathways (Appendix A). The top 10 enriched downregulated pathways included those for graft-versus-host disease, allograft rejection, systemic lupus erythematosus, and complement and coagulation cascades. The other six pathways were non-homologous end-joining, DNA replication, mismatch repair, protein export, malaria, and the Hedgehog signaling pathways. These results suggest that fatty acid biosynthesis and immune-related pathways may play important roles in RIF occurrence and progression.

### 3.4. Correlation Analysis of Signaling Pathways and Hub Genes

Differences in signaling pathways between RIF and healthy controls were evaluated using the ssGSEA algorithm for the GSE111974 dataset. Based on these results, the Spearman correlations between lipid metabolism pathways and immune pathways were analyzed (Figure 4A). The results showed that fatty acid biosynthesis was significantly negatively correlated with immune pathways (*p* < 0.05). There was no significant correlation between ovarian steroidogenesis, steroid hormone biosynthesis, and immune signaling. Fatty acid biosynthesis was significantly positively correlated with ovarian steroidogenesis and steroid hormone biosynthesis (*p* < 0.001). Ten hub genes were significantly correlated with lipid metabolism and immune pathways (*p* < 0.05) (Figure 4B); specifically, *RAB32, TRIB2, FAM155B,* and *SLC25A29* showed a higher correlation with fatty acid biosynthesis. Moreover, fatty acid biosynthesis had the highest enrichment score among all the significantly enriched signaling pathways (Appendix A). These findings indicate the importance of the fatty acid biosynthesis upregulation and downregulation of immune-related pathways in RIF, providing a reference for exploring RIF.

### 3.5. Identification of Hub Gene Expression Levels and Diagnostic Value

Hub gene expression levels were identified using box plots. In both the GSE111974 (Figure 5A) and GSE92324 datasets (Figure 5B), *NRCAM, ATP13A4, SHROOM3, FAM155B, ZNF436, SLC25A29, PDZD8,* and *HOXB3* had significantly higher expression levels in RIF than in healthy controls (*p* < 0.05), while *TRIB2* and *RAB32* expression levels were significantly reduced in RIF (*p* < 0.05). ROC analysis was executed to calculate the area under the curve (AUC) values to compare the sensitivity and specificity of hub genes in diagnosing RIF. The AUC value of every hub gene was greater than 0.600 in both datasets (Figure 6A,B). The combined diagnostic AUC value of the ten hub genes was 1.000 in both datasets (Figure 6C,D). Using the diagnostic nomogram, a combined diagnostic model of ten hub genes was constructed from the GSE111974 dataset (Figure 6E).

### 3.6. Prediction of the Underlying Mechanism of Hub Genes

To explore the underlying mechanism of hub genes, single-gene GSEA was performed. The significantly enriched top 5 upregulated and top 5 downregulated signaling pathways are presented (*p* < 0.05) (Appendix A). The high expression of eight hub genes was mainly associated with the upregulation of fatty acid biosynthesis (*ATP13A4, HOXB3, PDZD8, SLC25A29,* and *ZNF436*), thiamine metabolism (*ATP13A4, HOXB3, NRCAM, PDZD8,* and *ZNF436*), mineral absorption (*ATP13A4, SHROOM3, SLC25A29,* and *ZNF436*), steroid biosynthesis (*HOXB3, NRCAM,* and *PDZD8*), propanoate metabolism (*FAM155B* and *SHROOM3*), butanoate metabolism (*FAM155B*), 2-oxocarboxylic acid metabolism (*FAM155B* and *SHROOM3*), and glycosphingolipid biosynthesis-lacto and neolacto series (*HOXB3* and *SHROOM3*), and the downregulation of complement and coagulation cascades (*FAM155B, NRCAM, PDZD8, SLC25A29,* and *ZNF436*), graft-versus-host disease (*FAM155B, HOXB3, NRCAM, PDZD8, SHROOM3, SLC25A29,* and *ZNF436*), allograft rejection (*HOXB3, SLC25A29,* and *ZNF436*), DNA replication (*ATP13A4, FAM155B, SHROOM3, SLC25A29,* and *ZNF436*), and mismatch repair (*ATP13A4, FAM155B, SHROOM3, SLC25A29,* and *ZNF436*). The low expression of *TRIB2* and *RAB32* was associated with upregulated fatty acid biosynthesis, thiamine metabolism, and glycosphingolipid biosynthesis-lacto and neolacto series, and the downregulated complement and coagulation cascades, graft-versus-host disease, DNA replication, and mismatch repair. The single-gene GSEA results were consistent with changes in the expression of hub genes and the regulation of signaling pathways in RIF. The single-gene GSEA results of ten hub genes were in concordance with the GSEA results of all genes in the GSE111974 dataset, not only indicating that the ten hub genes are representative in RIF, but also implying that further investigation of the ten hub genes and the lipid metabolism and immune-related pathways may be necessary.

### 3.7. PPI and Correlation Analyses of Hub Genes and ERA

The betweenness centrality gradually increased from the outer circle to the inner circle. Two downregulated (*TRIB2* and *RAB32*) and eight upregulated (*NRCAM, ATP13A4, SHROOM3, FAM155B, ZNF436, SLC25A29, PDZD8,* and *HOXB3*) hub genes were shown in green and red, respectively. The 238 ERA genes were shown in yellow. In the PPI network, ten hub genes as the potential RIF biomarkers interacted with the ERA genes (Figure 7). In the correlation analysis, ten hub genes, specifically *RAB32, FAM155B,* and *SLC25A29,* showed significant correlation with most ERA genes (*p* < 0.05), including *NR4A2, SLC16A6, MAP2K6, CAPN6, DUOX1, RANBP17, BIRC3, COTL1, STEAP4,* and *PROM1* in the GSE111974 (Appendix A) and GSE92324 (Appendix A) datasets. The 10 ERA genes belonged to the intersection (Appendix A) of the 85 DEGs (Appendix A) and 238 ERA genes (Appendix A) and were enriched in the biological process network (Figure 3B) and NF-κB and TNF signaling pathways (Figure 3D). The expression trend of 10 ERA genes in RIF in both datasets was exactly the opposite of that in ERA (Appendix A). Considering the guiding significance of ERA, to a certain extent, this indicates that endometrial receptivity may be impaired in RIF. Given the interactions and correlations between hub genes and ERA, the impaired endometrial receptivity may be linked to hub genes. Hence, the ten hub genes may have a certain clinical diagnostic value and serve as complementary genes for ERA.

### 3.8. Immune Cell Infiltration and Its Correlation with Hub Genes

Given the potential roles played by immune signals in RIF, the differences in immune cell infiltration between RIF and healthy controls were evaluated using ssGSEA in the GSE111974 dataset (Figure 8A). In Natural killer T, macrophage, immature dendritic, CD56dim natural killer, activated dendritic, Th2, Th1, T follicular helper, regulatory T, immature B, effector memory CD8 T, central memory CD8 T, activated CD8 T, and activated CD4 T cells, a total of 14 immune cells showed significantly lower infiltration in the RIF group than in the controls (*p* < 0.05). Notably, the ratio of Th1 to Th2 cells did not change significantly between the RIF group and the healthy controls (1.089 vs. 1.073, *p* = 0.439). The ratio of Th17 to Tregs increased significantly between the RIF group and healthy controls (0.702 vs. 0.600, *p* = 0.007). In the GSE92324 dataset, 17 immune cells showed significantly reduced infiltration in RIF (*p* < 0.05) (Figure 8B). Significant correlations were observed among the 28 immune cells (*p* < 0.05) (Appendix A). Ten hub genes were significantly correlated with immune cells (*p* < 0.05) (Figure 8C).

### 3.9. Validation of Hub Genes in Human Endometrial Tissues

To further validate the important role of ten hub genes in RIF, we collected the endometrial tissues from the RIF and control women. RT-qPCR results (Figure 9) showed that *RAB32* and *TRIB2* expression levels were significantly lower in the RIF patients than in the controls, while the expression levels of *FAM155B* was significantly higher, consistently in both datasets. However, *PDZD8, SHROOM3,* and *SLC25A29* showed opposite results; their expression levels were significantly decreased in the RIF patients. The expression of another four hub genes (*ATP13A4, HOXB3, NRCAM,* and *ZNF436*) was not significantly different between the two groups. Hub genes were expressed in the endometrium. Detecting endometrial hub genes as biomarkers to predict RIF may be feasible in assisted reproductive technology.

## 4. Discussion

Implantation is a special type of inflammatory and immune process. Disturbance of the endometrial immune microenvironment may contribute to RIF development. An in-depth understanding of the detailed pathogenesis of RIF is urgently needed to develop new diagnostic and therapeutic strategies. Herein, we carefully and objectively explored the potential biomarkers and underlying biological signaling pathways in RIF.

*RAB32* and *TRIB2* had significantly lower expression levels in RIF, whereas *NRCAM, PDZD8, ZNF436, HOXB3, SLC25A29, SHROOM3, ATP13A4,* and *FAM155B* expression was significantly upregulated. Notably, in the hub gene analysis, considering that the GSE111974 and GSE92324 datasets came from different platforms and the selection of the RIF cases and control groups in the two datasets was different, we chose to analyze DEGs between the RIF cases and controls in the two datasets separately [24,25], rather than directly merge their original data for analysis [26]. ROC analyses validated that the hub genes may predict RIF with excellent specificity and sensitivity. The combined diagnostic AUC value of ten hub genes was 1.000 in both datasets, suggesting a better prediction effect of hub genes for RIF in both natural and COS cycles. RT-qPCR results further validated *RAB32, TRIB2,* and *FAM155B* expression trends in the endometrium in RIF women. Located in the endoplasmic reticulum (ER), mitochondria, and lysosomes, GTPase Rab32 (*RAB32*) modulates mitochondria-associated membrane properties [27] and cellular metabolism [28]. In the liver, the interaction between mitochondria and lipid metabolism induced by a high-fat diet is mediated through Rab32 [29]. As master regulators of DCs and macrophages, Rab proteins regulate the host defense response [30,31,32,33]. Tribbles (Trib) are kinase-like proteins [33]. In inflammatory bowel disease, the active inflamed tissue exhibits reduced Trib2 (*TRIB2*) expression, and Trib2 inhibits NF-κB activation mediated by Toll-like receptor 5 (TLR5) [34]. Tribbles also participate in fatty acid metabolism [35]. However, a previous study showed that *TRIB2* expression is upregulated in RIF, contrary to our results [36]. In a recent study, Zhang et al. [37] performed transcriptomic analysis of endometrial receptivity in Chinese women; the results revealed that *FAM155B* could be one of the endometrial receptivity biomarkers. Individual heterogeneity may account for the inconsistent expression of another seven hub genes in RIF in both datasets and RT-qPCR. In addition, the small sample size is another limitation of this study. Future studies with larger sample sizes are needed.

GO analysis demonstrated that DEGs in RIF were primarily enriched in lipid metabolism and immune processes. In KEGG analysis, the NF-κB, TNF, and adipocytokine signaling pathways were linked together, which is consistet with similar RIF studies showing dysregulated NF-κB and TNF pathways [24,38,39,40]. GSEA results demonstrated that RIF was significantly enriched in downregulated graft-versus-host disease, allograft rejection, complement and coagulation cascades, antigen processing and presentation, upregulated fatty acid biosynthesis, ovarian steroidogenesis, and steroid hormone biosynthesis. Downregulated immune pathways were also observed in RIF women under the COS cycle [22]. The results of our study indicate that upregulated lipid metabolism pathways and correlations between dysregulated immune and metabolic pathways also exist in RIF. The study by Ahmadi et al. [39] showed that steroid hormone biosynthesis was a potentially disrupted pathway in RIF patients. According to the Spearman correlation coefficient, fatty acid biosynthesis was significantly negatively correlated with immune pathways and significantly positively correlated with ovarian steroidogenesis and steroid hormone biosynthesis. This revealed that fatty acid biosynthesis might play an important role in disorders of the endometrial immune microenvironment. Research has shown that uterine receptivity and embryo implantation are closely related to lipid metabolism [41]. Substance synthesis in the human body is tightly controlled, including prostaglandin (PG) synthesis. PGs are key for successful embryo implantation [42,43]. Excess polyunsaturated fatty acids (PUFAs) can alter the expression and concentration of a variety of related enzymes and act as substrates and competitive inhibitors of epoxidation, affecting PG production [44]. During the follicular phase, PUFAs enlarge the follicles to increase steroid hormone synthesis [45]. Thus, the effects of increased fatty acid biosynthesis on RIF cannot be ignored.

Then, we explored the underlying mechanisms of ten hub genes, as their relationships with RIF were rarely reported. In single-gene GSEA, the high expression of eight hub genes and the low expression of two hub genes were primarily associated with upregulated metabolic pathways and downregulated immune pathways. This was consistent with the GSEA results of the GSE111974 dataset, which further illustrated that hub genes have a certain representation in RIF, and their associated signaling pathways deserve further study. 

In PPI and correlation analyses, hub genes exhibited complex interactions and significant correlations with most ERA genes. *RAB32* and *FAM155B* showed stronger correlations. This suggested that hub genes might be complementary genes of ERA and have certain clinical significance. Exploring hub genes and NF-κB and TNF signaling pathways is necessary. Moreover, endometrial transcriptomic analysis of Chinese women uncovered 68 DEGs as endometrial receptivity biomarkers, including 13 genes overlapping with ERA and 55 non-overlapping genes, including *FAM155B* [37]. GO and KEGG analyses of 68 DEGs revealed primary enrichment in immune response and arachidonic acid metabolism [37], further emphasizing the importance and significance of hub genes and their associated signaling pathways in RIF. Besides, the biological processes of 238 ERA genes also included immune response [23]. Given the unclear and contradictory conclusions of previous studies regarding the ERA [46,47,48], the PPI analysis in this study played a role in showing how ERA and hub genes complement each other.

Endometrial immune cells exert immunomodulatory functions that depend on their relative abundance and phenotypes, and they can be altered in response to external and internal environmental signals (i.e., nutrition, metabolism, infection, inflammation, microbial dysbiosis, stress, and developmental programming) [4]. The infiltration of 28 immune cells in the RIF endometrium was characterized herein using ssGSEA. The results showed significantly reduced immune cell infiltration, including for the frequently studied Tregs, Th1, Th2, CD56dim natural killer, macrophages, and dendritic cells. The ratio of Th1 to Th2 cells did not change significantly between RIF and healthy controls. Correlation analysis indicated that hub genes were involved in changes in immune infiltration. Reduced immune cell infiltration further validated the downregulation of immune pathways.

During WOI, P4 and E2 act on endometrial cells to stimulate the secretion of proinflammatory cytokines and chemokines, IL-6, IL-8, IL-15, and CXC chemokine ligands 10 and 11, used to activate and recruit numerous immune cells to the endometrium [49,50]. In healthy implantation, endometrial immune cells react to trophoblast alloantigens and then differentiate into tolerogenic and permissive cells, inseparable from various signals derived from trophoblasts [51,52,53]. However, if immune cells are reduced, display anomalous activity, or are insensitive to endometrial signals, proper immune adaptation and healthy implantation may not occur [4]. Therefore, we assume that the alteration of local immune cell infiltration in the endometrium may be closely related to RIF occurrence and development.

Uterine NK cells are predominantly CD56^bright^ CD16^−^ [11,54], and a minority of uNK cells are CD56^dim^ CD16^+^ [55]. CD56^bright^ cells mainly produce cytokines, whereas CD56^dim^ cells show more cytolytic activity and cytotoxicity than CD56^bright^ cells [56]. Giuliani et al. [57] thought that maintaining a low ratio of CD16^+^ to CD56^+^ NK cells may be important for establishing a successful pregnancy because of the proinflammatory nature of CD16^+^ NK cells [58]. We presume the reduced CD56^dim^ CD16^+^ NK cell infiltration may not be conducive to maintaining an inflammatory environment for embryo implantation.

In humans, enhanced or suppressed Th1-type immunity has been observed in the RIF endometrium [59]. The excessive increase and decrease in Th1 cells are both detrimental to embryo implantation. Given that the ratio of Th1 to Th2 cells did not change significantly between the RIF group and the healthy controls, we speculate that reduced Th1 and Th2 infiltration may not be the main cause of RIF. A precise balance between Th1 (TNF-α), Th2 (IL-10), Th17 (IL-17), and Tregs is required for successful implantation [15,60]. CD4^+^ CD25^+^ Tregs regulate the function of Teffs and can maintain peripheral tolerance [61,62]. Thus, the marked reduction of Treg infiltration may break the immune balance between Th17 and Tregs, which in turn interferes with the maternal immune tolerance to the transferred embryo.

Tolerogenic DCs, macrophages, and Tregs are closely linked and affect each other [63,64]. The reduction in any one of these cells will affect other immune cell populations, eventually destroying immune homeostasis and disrupting normal embryo transfer process. Accordingly, the endometrium exhibits a complex immune network. The fine-tuning of the phenotypic balance within immune cell populations is essential for establishing endometrial receptivity to implantation [4].

In summary, we not only identified potential diagnostic biomarkers for RIF, but also offered preliminary insights into the correlations between lipid metabolism and immune pathways. Moreover, this study explored the reduced immune infiltration pattern in RIF. Therefore, it provides a novel perspective for RIF diagnosis and treatment. In the future, further confirmatory experiments in vivo and in vitro are required to verify our conclusions. 

## Figures and Tables

**Figure 1 biomolecules-13-00406-f001:**
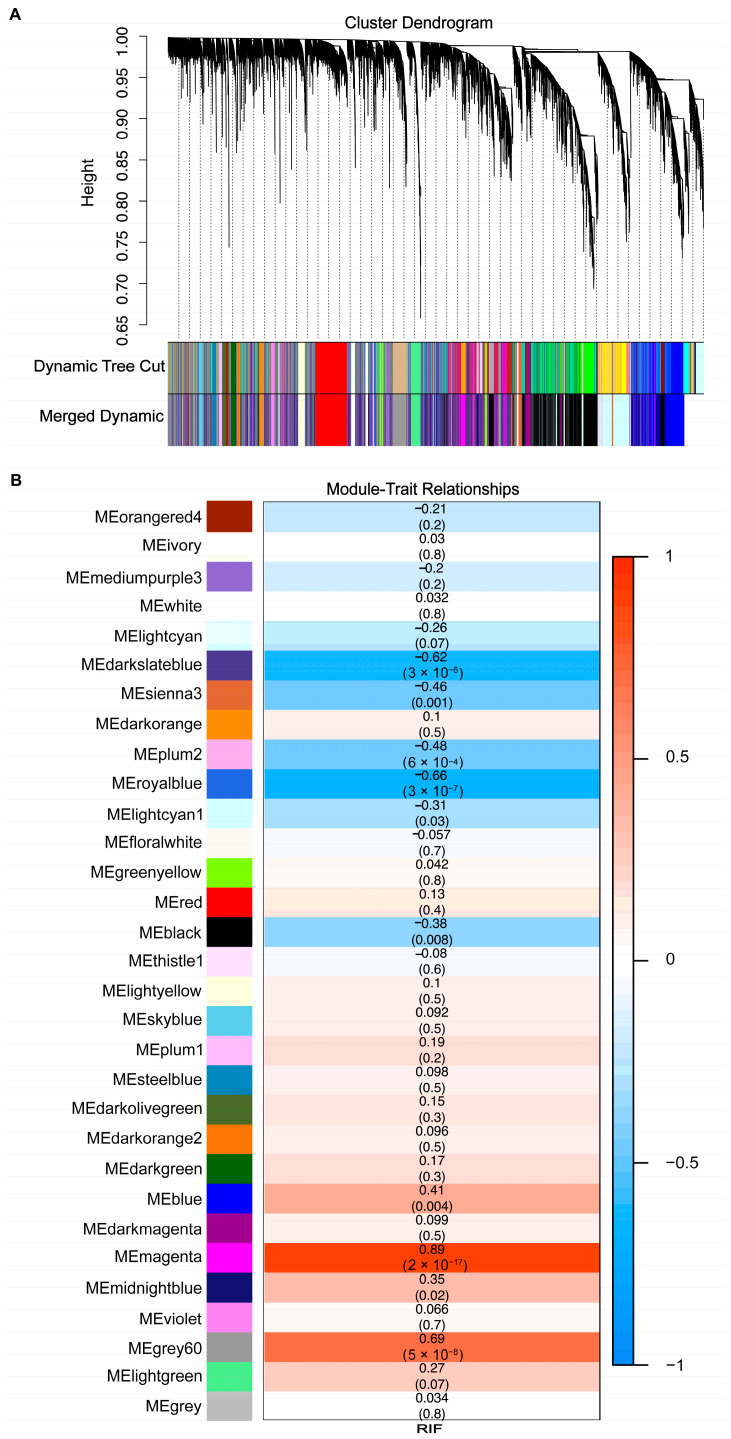
Construction of co-expression network and identification of hub modules. (**A**) Cluster dendrogram. Each branch indicates one gene, and each color represents one co-expression module. (**B**) Heatmap–module–trait relationships. The number in each module indicates the correlation coefficient and the corresponding *p*-value.

**Figure 2 biomolecules-13-00406-f002:**
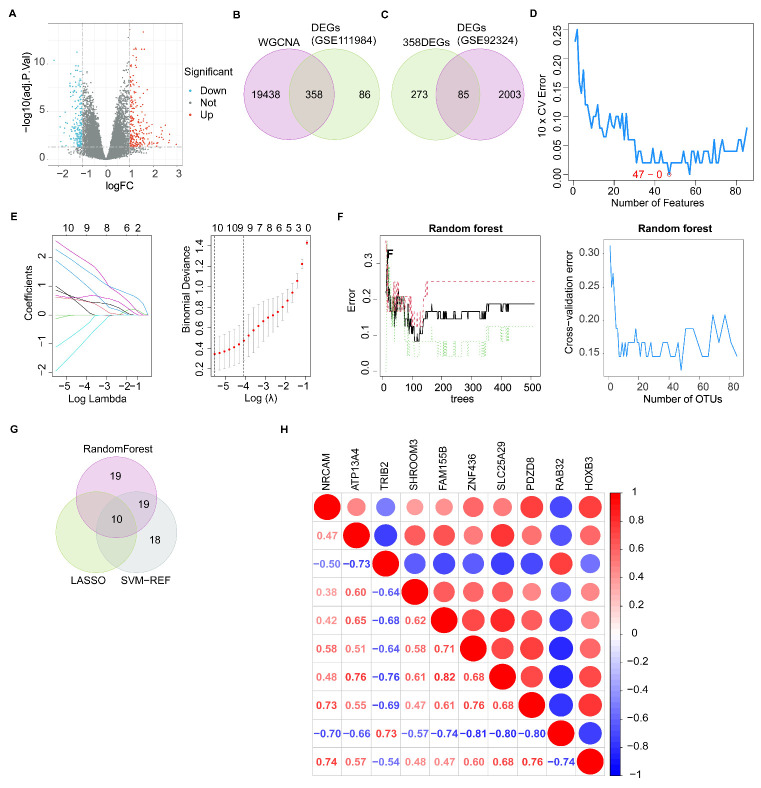
Identification of DEGs and screening of hub genes. (**A**) Volcano plot for DEGs in the GSE111974 dataset between RIF and healthy controls. (**B**,**C**) Venn diagram of preliminary gene screening. (**D**) SVM-RFE, (**E**) LASSO, and (**F**) RF analyses of the 85 DEGs. (**G**) Venn diagram of the ten hub genes. (**H**) Heatmap of the correlations between the hub genes. DEGs, differentially expressed genes; RIF, recurrent implantation failure; SVM-RFE, support vector machine-recursive feature elimination; LASSO, least absolute shrinkage and selection operator; RF, random forest.

**Figure 3 biomolecules-13-00406-f003:**
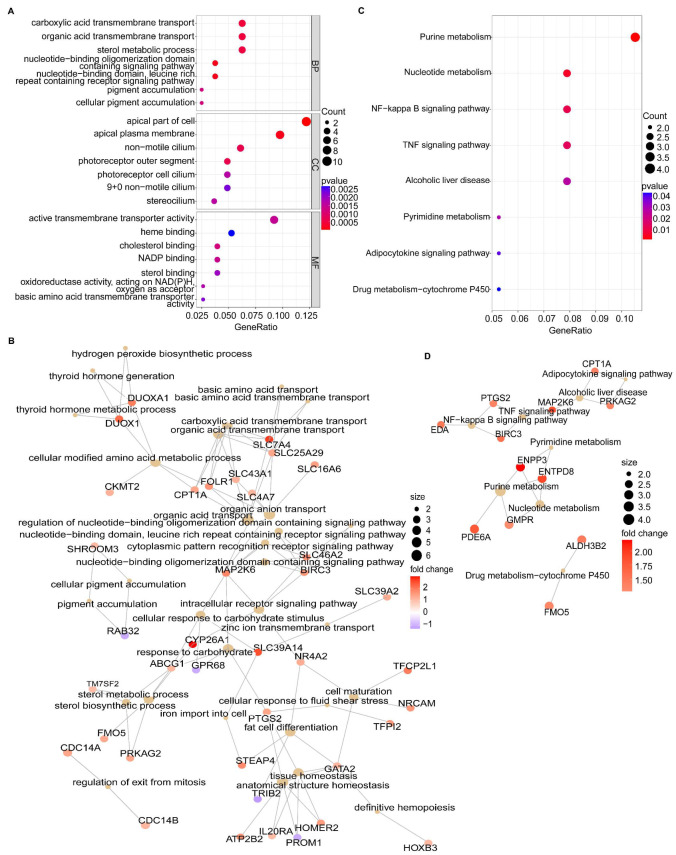
Functional enrichment analysis. (**A**) GO analysis of the 85 DEGs. (**B**) GO-cnet plot (the top 30 biological processes). (**C**) KEGG analysis of the 85 DEGs. (**D**) KEGG-cnet plot. GO, Gene Ontology; DEGs, differentially expressed genes; KEGG, Kyoto Encyclopedia of Genes and Genomes.

**Figure 4 biomolecules-13-00406-f004:**
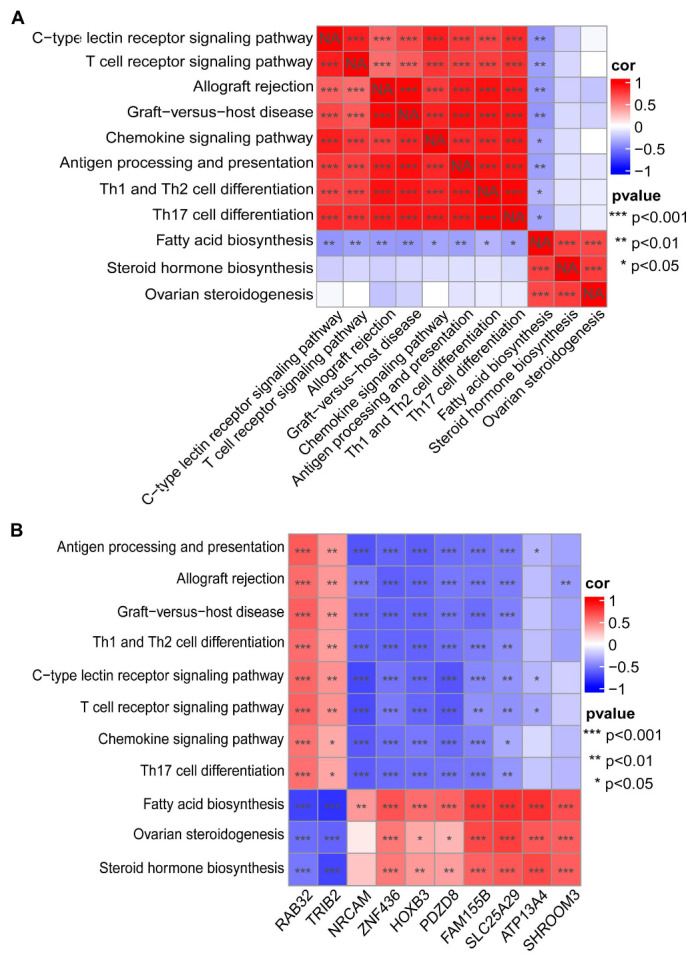
Correlation analysis of signaling pathways and hub genes. (**A**) Heatmap of the correlation between lipid metabolism and immune pathways. (**B**) Heatmap of the correlation between signaling pathways and the ten hub genes.

**Figure 5 biomolecules-13-00406-f005:**
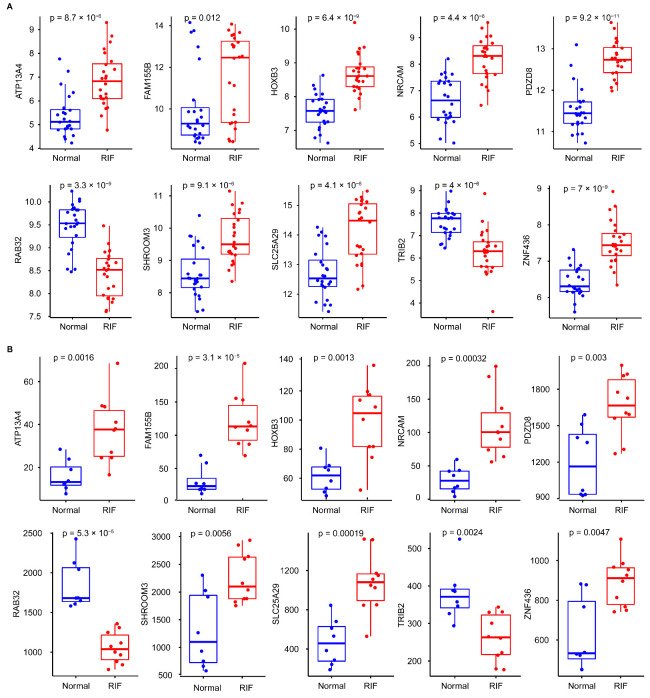
Identification of the hub gene expression levels. The ten hub genes in the (**A**) GSE111974 and (**B**) GSE92324 datasets. The unpaired Student’s *t* test was applied.

**Figure 6 biomolecules-13-00406-f006:**
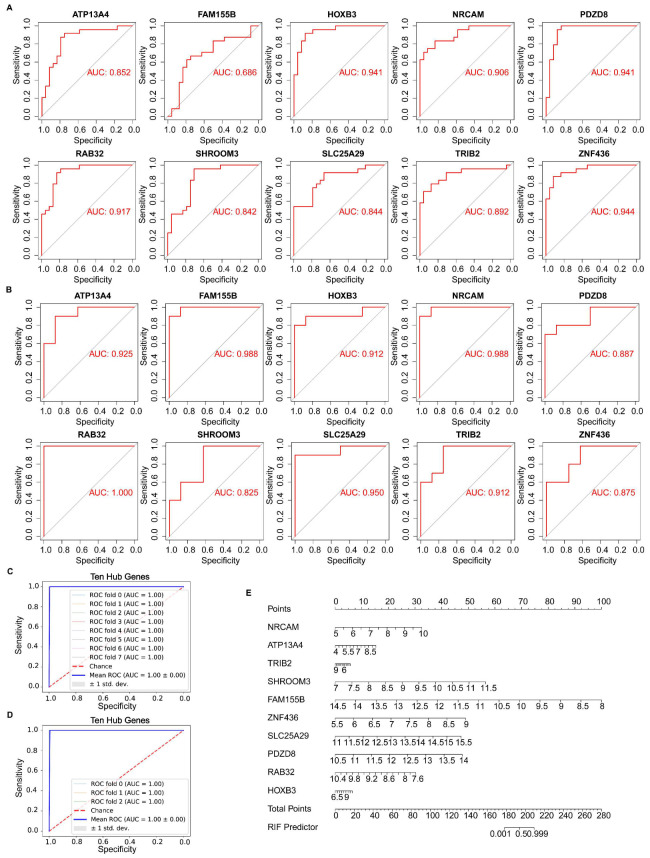
Validation of the hub gene diagnostic value. Every hub gene in the (**A**) GSE111974 and (**B**) GSE92324 datasets. Combined validation of the ten hub genes in the (**C**) GSE111974 and (**D**) GSE92324 datasets. (**E**) Diagnostic nomogram for the ten hub genes in the GSE111974 dataset.

**Figure 7 biomolecules-13-00406-f007:**
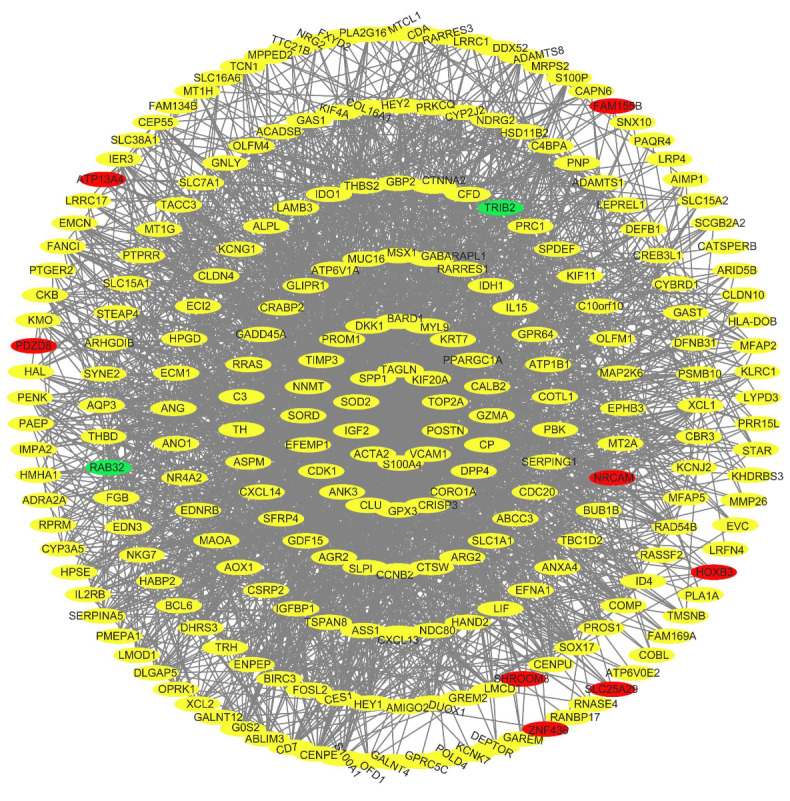
Protein–protein interaction network. Ten hub genes as potential RIF biomarkers interacted with ERA genes.

**Figure 8 biomolecules-13-00406-f008:**
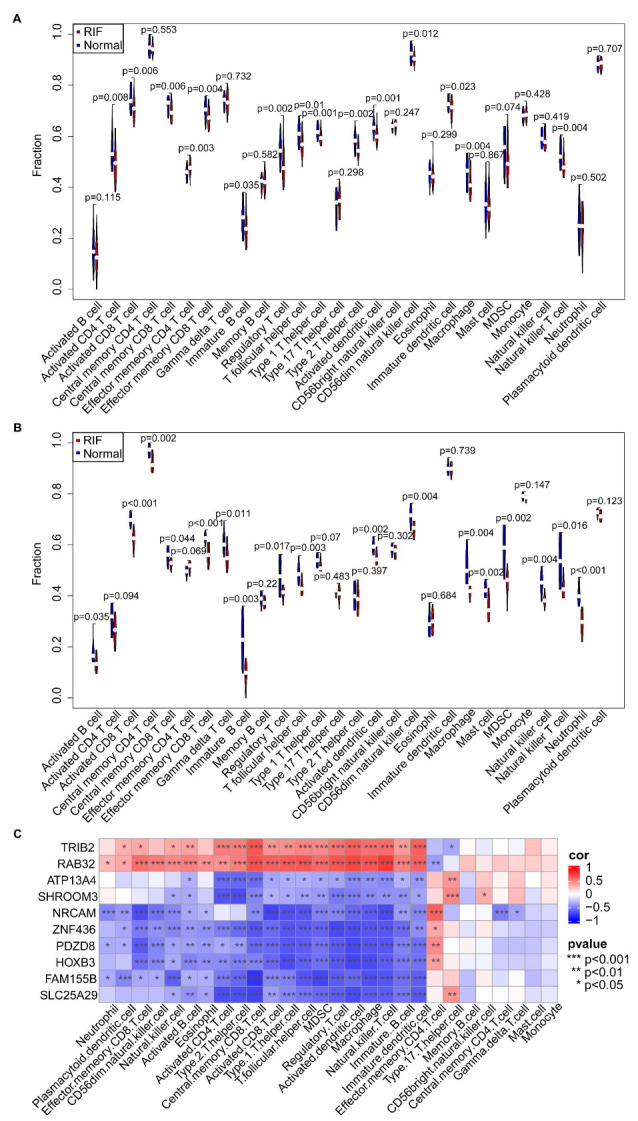
Landscape of immune cell infiltration. Violin plot of the 28 infiltrating immune cells in the (**A**) GSE111974 and (**B**) GSE92324 datasets. (**C**) Heatmap of the correlation between the ten hub genes and the 28 infiltrating immune cells in the GSE111974 dataset.

**Figure 9 biomolecules-13-00406-f009:**
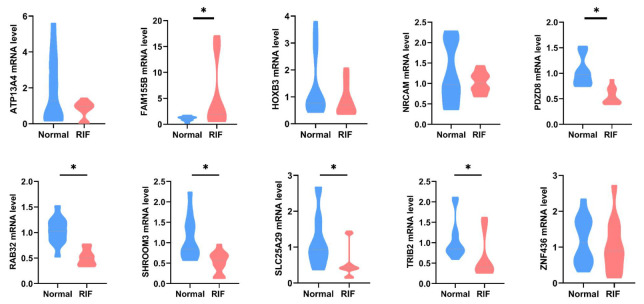
Validation of hub genes in human endometrial tissues. The unpaired Student’s *t* test was applied. * *p* < 0.05 indicates significant differences between RIF patients and healthy controls.

## Data Availability

GSE111974 and GSE92324 datasets are publicly available and were extracted from the online GEO repository (http://www.ncbi.nlm.nih.gov/geo/, accessed on 14 May 2022).

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
