# Peer review of "Potential Biomarkers and Endometrial Immune Microenvironment in Recurrent Implantation Failure"

_biomolecules, 2023, doi:10.3390/biom13030406_

Round 1

Reviewer 1 Report

The authors well described the gene expression profile of the RIF. RIF is a severe problem for reproductive-aged women. therefore, this manuscript will enhance the understanding of readers. 

Author Response

Responses to Reviewer 1

Comments 1: The authors well described the gene expression profile of the RIF. RIF is a severe problem for reproductive-aged women. therefore, this manuscript will enhance the understanding of readers.

Response 1: Thank you for your careful review and approval.

Reviewer 2 Report

This paper addresses the complex problem of multiple implantation failure observed in assisted reproductive procedures. The aim of the paper was to re-analyse gene expression in published databases that had been analysed in two other , published papers. However, the main set of data analysed were the differentially expressed gene bases obtained in the paper by Bast et al. 2019 

However, several important points come to my attention.

1. firstly, the authors collectively analysed results of  Basyu et al. and Pathare et al.("Genes from the candidate hub modules in WGCNA were intersected with the identified DEGs (GSE111974). To alleviate the instability of a single dataset and further improve the accuracy of results, the intersecting genes were intersected with DEGs from another dataset (GSE92324)." ) The selection of patients and control groups in these studies was different and, most importantly, the material in one experiment was taken from patients in their natural cycle and in the other from hormonally stimulated patients.

Therefore, gene expression in the two groups is not comparable. And this is a serious methodological flaw of this work 

In addition, the two studies used different expression matrices , which also did not allow for a combined analysis of the expression of the selected genes.

Therefore, calculations should be performed anew taking into account one or the other database. 

2 Despite repeated in-depth analysis, the overall conclusions in the reviewed paper and in the paper of Batu et al.  were not significantly different.

Both papers indicated differences in gene expression of the NF-kB and TNF-alpha pathways.

The indication in reviewed   paper of the metabolic pathway is additional information that was obtained from the analysis and should in principle only be discussed and presented as a new information.

3. there is a great deal of information in the paper about the importance of immune cells in the process of implantation ,. This information is well known and the conclusions of the paper indicating the necessity of Th1/Th2 balance are not new. It should be  substantially shortened or discarded 

4. description/characterisation of the patient groups from the cited papers is missing, which should absolutely be included in the chapter material and methods 

5. the paper makes little reference to other similar work on described similar  gene expression pathways  in RIF  

Author Response

Responses to Reviewer 2

Reviewer 3 Report

Review manuscript : Potential Biomarkers and Endometrial Immune Microenvironment in Recurrent Implantation Failure

I read with great interest the manuscript by Li and colleagues submitted to biomolecules. Recurrent implantation failure is a relevant issue in human assisted reproduction for which there isn’t an answer nor a treatment yet. In this scenario, new studies employing experimental and new analytical tools are relevant and results shall add important information to help couples suffering from the condition. The present study explores genome databases and performs real-time PCR to analyze the interaction of factors that may be present on implantation failure. I am not an expert on the use of in silico systems. However, I can understand the rationale of the methods and their potential for therapeutic uses. I consider the authors did an interesting study and their results add important information on new therapeutic approaches to RIF patients. The paper is well written and the Figures are clear and informative. I recommend the publication of this manuscript.

Author Response

Responses to Reviewer 3

Comments 1: I read with great interest the manuscript by Li and colleagues submitted to biomolecules. Recurrent implantation failure is a relevant issue in human assisted reproduction for which there isn’t an answer nor a treatment yet. In this scenario, new studies employing experimental and new analytical tools are relevant and results shall add important information to help couples suffering from the condition. The present study explores genome databases and performs real-time PCR to analyze the interaction of factors that may be present on implantation failure. I am not an expert on the use of in silico systems. However, I can understand the rationale of the methods and their potential for therapeutic uses. I consider the authors did an interesting study and their results add important information on new therapeutic approaches to RIF patients. The paper is well written and the Figures are clear and informative. I recommend the publication of this manuscript.

Response 1: Thank you for your careful review and approval.

Reviewer 4 Report

-introduction is to long, I suggest to left out everything from including ref 13 to ref 40, or at least very shorten this paragraph

- it was proven lately in several clinical studies that ERA doesn't work so I don't find any conclusions from this aspect relevant

- for samples vs. four controls is very low sample

- discussion is to long and reader lose focus very quickly, it should be significantly shorter 

Author Response

Responses to Reviewer 4

Round 2

Reviewer 2 Report

I would like to thank the authors of the paper for taking my comments into account and improving the manuscript 

The article has been revised and supplemented accordingly.

Readers with adequate knowledge of the source of the data in the context of patient selection will be able to critically assess the value of the data provided and the conclusions drawn from it

Author Response

Responses to Reviewer 2

Thank you for your careful review and approval. 

Reviewer 4 Report

The authors have substantially changed the manuscript according to my comments, but this doesn't change the fact the number of included samples in very low. 

Author Response

Comments: The authors have substantially changed the manuscript according to my comments, but this doesn't change the fact the number of included samples in very low. 

Response: Thank you very much for pointing out this important issue. We agree with your comment that a larger sample size is necessary. Your suggestion provides a direction for our next research. Unfortunately, more samples have not been collected due to the limited time and funding. In this study, we aimed to explore the potential biomarkers, underlying signaling pathways, and contribution of immune cell infiltration in RIF. Thus, we mainly focus on the results of bioinformatics analysis. Then, we revised the discussion and refined the limitations: “In addition, the small sample size is a limitation of this study as well. Future studies with larger sample sizes are needed.” (Lines 441–443).